# Melatonin Type 2 Receptor Activation Regulates Blue Light Exposure-Induced Mouse Corneal Epithelial Damage by Modulating Impaired Autophagy and Apoptosis

**DOI:** 10.3390/ijms231911341

**Published:** 2022-09-26

**Authors:** Rujun Jin, Ying Li, Hui Jin, Hee Su Yoon, Ji Suk Choi, Jonghwa Kim, Hyeon Jeong Yoon, Kyung Chul Yoon

**Affiliations:** 1Department of Ophthalmology, Chonnam National University Medical School and Hospital, Gwangju 61469, Korea; 2Department of Biomedical Sciences and Centers for Creative Biomedical Scientists at Chonnam National University, Gwangju 61469, Korea

**Keywords:** blue light, melatonin type 2 receptor, autophagy, apoptosis, corneal damage

## Abstract

The MT1/2 receptors, members of the melatonin receptor, belong to G protein-coupled receptors and mainly regulate circadian rhythms and sleep in the brain. Previous studies have shown that in many other cells and tissues, such as HEK293T cells and the retina, MT1/2 receptors can be involved in mitochondrial homeostasis, antioxidant, and anti-inflammatory responses. In our study, we aimed to investigate the effects of blue light (BL) exposure on the expression of melatonin and its receptors in the mouse cornea and to evaluate their functional role in corneal epithelial damage. After exposing 8-week-old C57BL/6 mice to BL at 25 and 100 J/cm^2^ twice a day for 14 days, a significant increase in the expression of 4-HNE and MT2 was observed in the cornea. MT2 antagonist-treated mice exposed to BL showed an increased expression of p62 and decreased expression of BAX and cleaved caspase 3 compared with mice exposed only to BL. In addition, MT2 antagonist-treated mice showed more enhanced MDA and corneal damage. In conclusion, BL exposure can induce MT2 expression in the mouse cornea. MT2 activation can modulate impaired autophagy and apoptosis by increasing the expression of BAX, an apoptosis activator, thereby regulating the progression of corneal epithelial damage induced by BL exposure.

## 1. Introduction

The anterior surface of the eye acts as a barrier to the external environment and protects the delicate underlying tissues from damage [1]. The corneal epithelium is a self-renewing stratified squamous epithelium that serves as the frontline of the innate immune system to protect the eye from microbial invasion [1,2]. Exposure to light emitting diode (LED) lights, the use of smartphones, and computer work could expose the ocular surface to light stress, which can increase apoptosis and damage in the epithelium. Our previous study showed a significant reduction in tear secretion and tear film break-up times, and increased inflammatory infiltration and apoptosis on the ocular surface of mice or humans exposed to LED lights or smartphones for extended periods of time [3,4,5].

Oxidative stress is associated with the pathogenesis of many ocular diseases, including age-related macular degeneration disease, glaucoma, and dry eye. Mitochondrial dysfunction is linked to oxidative stress and reactive oxygen species (ROS) [6,7,8]. Additionally, previous studies have demonstrated that light-stress could also induce oxidative damage [9,10]. The levels of 4-hydroxynonenal (4-HNE) and malondialdehyde (MDA), indicators of oxidative-stress-induced lipid peroxidation products, are significantly increased in oxidative stress-induced damage [6,9,10,11].

Melatonin, a hormone secreted by the pineal gland, regulates circadian rhythm, including sleep and other cyclical bodily activities [12]. The action of circulating melatonin is mediated by two main pathways, receptor-dependent and non-receptor-dependent, both inhibiting adenylate cyclase via Gi protein, thereby reducing the cAMP content [12,13,14,15]. Among other actions, melatonin and its receptors (MT1/2) are involved in the antioxidation, anti-inflammation, and regulation of mitochondrial homeostasis and are related to the redox status of cells and tissues [16,17,18,19]. In addition, recent studies have found high concentrations of extrapineal melatonin production in damaged tissues and cancer cells, and have shown that local elevations of melatonin are involved in apoptosis, antioxidant, and anti-inflammatory responses [12,15]. MT1 and MT2 receptors belong to the guanine nucleotide-binding regulatory protein-coupled receptor family and respond to a variety of extracellular stimuli, including hormones, neurotransmitters, and growth factors [14]. MT1/2 expression may be involved in several regulatory processes, including cellular injury and apoptosis process, by mediating other signaling pathways, such as mitogen-activated protein kinase and extracellular-signal-regulated kinase (ERK1/2) pathways [20,21,22]. Interestingly, a recent study has shown that melatonin inhibits apoptotic cell death induced by Vibrio vulnificus VvhA via MT2 coupled with neutrophil cytosolic factor-1 [23].

In the eye, MT1 and MT2 are found in the retina, lens, and cornea [12,15,24,25]. Both MT1 and MT2 are differentially distributed throughout the retina, and the expression of MT1 in dopaminergic amacrine cells could directly inhibit dopamine release [16,26]. In addition, MT1 expression in the retina is involved in the bioluminescence circadian rhythm in mice [22].

A recent study showed that MT2 in the corneal epithelium may be involved in corneal wound healing by regulating the rate of migration [27]. However, the expression of melatonin receptors in the cornea and their effects are not well understood. In the present study, we aimed to investigate whether blue light (BL) exposure in mice can trigger the activation of melatonin receptors in corneal epithelial cells, and further investigated the role of melatonin receptors in light-induced corneal apoptosis.

## 2. Results

### 2.1. Oxidative Stress Levels in the Cornea Induced by BL Exposuree

In this study, changes in the oxidative stress levels of 4-HNE (Figure 1A) and MDA (Figure 1B,C) were detected during 14 days of BL exposure. The 25 J and 100 J BL exposure groups showed higher levels of 4-HNE and MDA in the mouse cornea than the untreated (UT) group (all *p* < 0.01). When comparing MDA levels between days 3, 7, and 14, mice exposed to 25 J and 100 J BL showed significantly higher levels of MDA at day 14 (Figure 1B; all *p* < 0.01). Mice exposed to 100 J BL showed increased MDA levels from day 3 of induction (Figure 1B; all *p* < 0.05). 4P-PDOT (N-(1,2,3,4-Tetrahydro-4-phenyl-2-naphthalenyl) propenamide) was a selective and affinity MT2 antagonist as a protein. 4P-PDOT can significantly counteract melatonin-mediated antioxidant effects and inhibit the expression of BAX induced by MT2 activation. In our results, MT2 antagonist-treated mice had significantly increased MDA levels (*p* < 0.01) compared with mice exposed only to BL at day 14 (Figure 1C). Our results showed that BL exposure induced oxidative stress in the cornea. A significant increase in oxidative stress in the MT2 antagonist-treated groups suggested that MT2 receptors may be involved in oxidative stress in corneal epithelial cells.

### 2.2. Melatonin Secretion and MT1/2 Activation in Corneal Tissues Due to BL Exposure

The expression of the MT1 and MT2 receptors was observed using Western blotting (Figure 2, Figure 3 and Figure 4) and IHC (Figure 2, Figure 3 and Figure 4). Magnified images of the representative corneal sections stained with MT1 and MT2 (green), and counterstained with 40,6-diamidino-2-phenylindole (DAPI) (blue), are presented in Figure 2, Figure 3 and Figure 4. MT1 was expressed in the corneal stroma, and MT2 was expressed in the corneal epithelium and endothelium (Figure 2A). In addition, there was no significant difference in MT1 expression between the UT, 25 J BL, and 100 J groups. However, mice exposed to 25 and 100 J BL showed a more increased expression of MT2 in the cornea compared with UT (Figure 2B).

Melatonin levels were detected using ELISA, as shown in Figure 3. Our results showed that mice exposed to 25 J and 100 J BL had increased levels of melatonin secretion in the cornea (Figure 3A,C; all *p* < 0.01) and MT2 expression (Figure 3B,D–F; all *p* < 0.01) compared with UT mice. Mice exposed to 25 J BL showed a reduced melatonin expression at 14 days compared with at 3 and 7 days (Figure 3A; all *p* < 0.01). Mice exposed to 100 J BL showed decreased melatonin levels on days 7 and 14 (Figure 3C; all *p* < 0.01). Conversely, the expression of MT2 was increased in mice exposed to 25 J BL on day 14 (Figure 3B,E,F; all *p* < 0.01). The group exposed to 100 J showed significantly greater differences in MT2 expression at days 7 and 14 compared with day 3 and UT (Figure 3D–F; all *p* < 0.01). In addition, mice treated with MT2 antagonists showed increased levels of melatonin production compared with the untreated mice, after exposure to either 25J or 100J BL for 14 days (Figure 4A; *p* < 0.05). However, MT2 expression did not increase in MT2 antagonist-treated mice, as shown in Figure 4B,C. The MT1 expression levels were not significantly different between the UT, BL exposed (25 J and 100 J) groups, and antagonist-treated groups (Figure 4B).

### 2.3. Autophagy Flux Defects in the Mouse Cornea Due to BL Exposure

Autophagy begins with the formation of a phagophore, a crescent-shaped double membrane closely linked to LC3II, and p62 was the first selected autophagy adapter found in mammals [28]. However, the overexpression of p62 had a protective effect on cell survival and inhibited autophagic influx. Many studies have demonstrated that autophagy was responsible for the degradation of p62. Thus, impairment of autophagy was usually accompanied by a massive accumulation of p62, followed by the formation of aggregate structures positive for p62 [28,29,30,31].

The Western blot analysis showed that the expression of LC3-II, an autophagosome marker, was significantly increased in the cornea after exposure to 25 J and 100 J BL compared with UT (Figure 5A,B, *p* < 0.01). MT2 antagonist-treated mice showed more increased LC3-II expression than BL-exposed mice (Figure 5C; *p* < 0.01). We found that the expression of p62, a marker of lysosomal degradation, increased after exposure to 25 J and 100 J BL (Figure 5A,B; *p* < 0.01). In addition, mice treated with the MT2 antagonist showed a greater increase in p62 expression than mice treated with 25 J and 100 J BL alone (Figure 5C; *p* < 0.01), suggesting that the MT2 receptor was involved in the BL exposure-induced autophagy response. IHC for p62 expression are shown in Figure 5D,E.

### 2.4. BAX and Cleaved Caspase 3 Expression in Corneal Tissue induced by BL Exposure

Bcl-2-associated X (BAX), a member of the Bcl-2 family, is an important regulator of apoptosis [32]. As a result of evaluating the change in BAX expression after exposure to BL, similar to the change in MT2 expression, the mice exposed to BL at 25 J showed an increased BAX expression in the cornea on day 14 (Figure 6A; *p* < 0.01). In addition, the 100 J BL exposure group showed a greatly enhanced expression of BAX on days 3, 7, and 14 compared with the UT group (Figure 6B; *p* < 0.01). In contrast, we demonstrated that the expression of BAX and cleaved caspase 3 in mice treated with MT2 antagonists was significantly reduced in mice exposed to 25 J or 100 J BL only (Figure 6C; *p* < 0.01). Our results indicated that MT2 receptors could modulate the expression of BAX and cleaved caspase 3 in the cornea under BL exposure.

### 2.5. Corneal Epithelial Damage

Mice exposed to 100 J BL with or without the MT2 antagonist showed an increase in corneal fluorescein staining scores (CFS) at 14 days compared with UT, and the antagonist-treated group had a greater increase in corneal damage than the 100 J BL exposure group (Figure 7; all *p* < 0.01). Although no increase in damage was found in the 25 J BL exposure group, mice exposed to 25 J BL and treated with the MT2 antagonist had increased corneal damage on day 14 compared with the mice exposed to UT and 25 J BL (Figure 7; both *p* < 0.01). This suggests that MT2 receptor activation may protect against corneal epithelial damage caused by BL exposure.

## 3. Discussion

Melatonin, the “night clock”, released by the pineal gland, is maintained at low levels in C57BL/6 mice and may not be involved in circadian rhythms and sleep regulation [33,34]. However, there is growing evidence that the local production of melatonin is found in other organs, tissues, and cells including the retinal pigment epithelium, lens, and lymphocytes, and may be involved in anti-inflammatory and antioxidative responses [17,35,36]. In addition, some previous studies have shown that the expression of MT1/2 receptors was found in C57BL/6 mice and has an important role in ROS homeostasis, mitochondrial protection, movement sensitization, and apoptosis [33,34,35]. However, little is known about the function of melatonin and its receptors changes, and their antioxidant effects in ophthalmic diseases. Our findings may explain the mechanism through which the local production of melatonin and its receptor activation act on impaired autophagy induced by BL exposure.

The autophagy process is generally recognized as playing an essential role in the homeostasis of organisms by mediating the breakdown and transformation of defective organelles within cells [37,38]. While there is evidence that the induction of autophagy promotes cell survival under several types of stress, there is also a growing body of research showing that autophagy overreaction can promote cell and tissue damage [38,39,40]. p62 serves as a central adapter for the mammalian target of rapamycin complex 1 (mTORC1) activation at the lysosomal surface, and it has been proposed to contribute to selective autophagy of protein aggregates (aggrephagy), depolarized mitochondria (mitophagy), and invasive microbes (xenophagy) through ubiquitin-signaling [29,41,42]. Moreover, many previous studies have shown that the abnormal accumulation of p62-positive aggregated structures is detected in patients with liver disorders, tumors, and neurodegenerative diseases [28,29,30,31]. Although some previous studies have evaluated autophagy markers in light-induced stress injury, the precise changes in autophagic flux and its underlying regulatory mechanisms remain unclear. In this study, we found an increased expression of LC3-II and p62 after BL exposure, with or without MT2 antagonist treatment. Mice exposed to low-intensity BL showed a significantly reduced expression of p62, a marker of lysosomal interpretation, on days 3 and 7. However, p62 expression was significantly increased 14 days after BL stimulation. In addition, in mice exposed to high-intensity BL, a significant increase in p62 expression was observed from day 7. These results suggest that corneal epithelial cells suffer a defect in autophagic flux after exposure to BL, and that prolonged BL exposure, even at a low intensity, could induce impaired autophagy.

Reactive oxygen species (ROS) are well known to be key intracellular signaling molecules, and the overproduction of ROS leads to impaired mitochondrial function [43]. Oxidative stress is one of the stimuli contributing to the induction of ROS, which can induce cellular autophagy. However, previous studies have shown that ROS-induced mitochondrial damage can further promote impaired autophagy, which can lead to increased intracellular and extrinsic ATP imbalance, oxidative stress, and tissue damage [40,44]. In the present study, mice with and without MT2 antagonists under BL stimulation showed an increased expression of LC3-II and p62. However, the expression of LC3-II and p62 was further enhanced in MT2 antagonist-treated mice. Although melatonin secretion was increased in the MT2 antagonist-treated groups compared with the untreated groups, it was found that MDA increased more in the corneas of mice treated with the MT2 antagonist. Taken together, our results show that prolonged exposure to BL enhances the aberrant autophagy response and reduces lysosomal interpretation owing to excessive ROS release. Furthermore, this suggests that an increase in impaired autophagy may lead to a vicious cycle of further increasing tissue oxidative stress. Controlling autophagy and apoptosis is important for reducing corneal damage.

Melatonin and its receptors, MT1 and MT2, are involved in antioxidation, anti-inflammation, and the regulation of mitochondrial homeostasis [16,38,45]. Although previous studies have shown that melatonin protects cells by increasing cell proliferation and redirecting glucose oxidation to the mitochondria, the functions of melatonin and its receptors in the eye remain unclear. In this study, we observed that MT2 was located on the corneal epithelium and endothelium. We also found that BL could induce an increase in melatonin secretion, which protects corneal epithelial cells from oxidative stress and promotes normal autophagy. Nevertheless, long-term BL stimulation locally reduced melatonin production in mouse corneas. We also found that a significant increase in the activation of MT2 in the corneal epithelium promoted the expression of BAX, a pro-apoptotic factor, thereby decreasing the abnormal autophagy response and damage in the mouse cornea. In contrast, the corneas of the mice treated with the MT2 antagonist showed a markedly impaired autophagy response and MDA. These findings suggest that melatonin secretion and the activation of MT2 can protect cells from ROS stress and reduce corneal epithelial damage by regulating autophagy and apoptosis via BAX expression. Although MT2 activation promotes apoptosis under BL-induced stress, our current data indicate that MT2 induction protects corneal epithelial cells from impaired autophagy and ROS stress. Many previous studies have demonstrated that apoptosis is an important pathway that protects cells from damage, as the balance between BCL2 and BAX can protect mitochondrial function and proliferation in normal cells and other tissues [4,46,47,48,49,50].

After BL exposure, mice not treated with the MT2 antagonist showed increased MT2 expression, but decreased melatonin secretion, compared with the mice treated with the MT2 antagonist. We believe that the increased secretion of melatonin in the antagonist-treated group could be attributed to the suppression of MT2 expression, suggesting that melatonin increased so as to directly control oxidative stress and promote autophagy response. However, additional studies including the effects of MT2 expression and MT2 antagonist treatment on the laser or chemical burn-induced mouse cornea may be necessary.

In conclusion, BL exposure induces MT2 activation. MT2 expression can reduce abnormal autophagy responses by increasing the activation of BAX (Figure 8). Our study suggests that the MT2-BAX signaling pathway may be involved in impaired autophagy and apoptosis induced by BL exposure, ultimately regulating corneal epithelial damage.

## 4. Materials and Methods

### 4.1. Designs of Mouse Model and Experiments

The research protocol was approved by the Chonnam National University Medical School Research Institutional Animal Care and Use Committee (CNUIACUC-20001). All the animals were treated in accordance with the ARVO Statement for the Use of Animals in Ophthalmic and Vision Research. Female C57BL/6 mice aged 8 weeks were used in the subsequent experiments. During these experiments, animal behavior, food intake, and water intake were not restricted. In addition, mice were allowed to acclimatize under standard laboratory conditions at Chonnam National University Hospital Veterinary Hospital 1 week prior to the initiation of the experiment, and maintained a temperature of 25 °C under 12-h contrast cycle illumination (bright: 08:00 a.m.– 20:00 p.m.; dark: 20:00 p.m.– 08:00 a.m.).

This study included a three-part (parts I, II, and III) experiment, and light induction was performed twice daily (irradiation began at 21:00 p.m. and 04:00 a.m. to avoid variation) for 14 consecutive days. In part I (n = 4), mice were exposed to BL at energy doses of 25 and 100 J/cm^2^ and were then divided into 25 J and 100 J groups. Mice in the UT group were not exposed to BL and served as untreated controls during the experiment. Mice in part II (n = 7) were irradiated with BL at an energy dose of 25 or 100 J/cm2 for 3, 7, or 14 days. In part III (n = 7), the mice were divided into five groups: UT, 25 J BL, 25 J BL with MT2 antagonist, 100 J BL, and 100 J BL with MT2 antagonist. The BL groups were induced with the respective energy doses for 14 days, and 2 μL of MT2 selective antagonist (4P-PDOT; catalog no. ab146419) was instilled into the antagonist-treated mice. The mice were confined in an adjustable retaining cage in a dark room, where the BL was placed 5 cm above and perpendicular to the head of the mouse; only the ceiling of the cage emitted light. Therefore, at any given instant, the dose of light on the mouse ocular surface depends on the head posture. We estimated that the mouse kept its head aligned with its body, on average, while in the retaining cage [3,4].

At the end of the experiment, the animals were sacrificed. Enzyme-linked immunosorbent assay (ELISA) for MDA and melatonin, and immunofluorescent staining for 4-HNE, melatonin type (MT)1/2 receptors, and p62 were performed. In addition, the expression of MT1, MT2, light chain 3-II (LC3-II), p62, BAX, and cleaved caspase 3 were evaluated by Western blotting after BL stimulation and CFS by slit lamp biomicroscopy.

### 4.2. ELISA

The total protein levels of MDA (Cell Biolabs, San Diego, CA, USA) and melatonin (catalog no. OKEH02566; Aviva Systems Biology, San Diego, CA, USA) were determined using ELISA (n = 3). Tissues were collected and pooled in a lysis buffer containing protease inhibitors for 30 min. The cell extracts were centrifuged at 14,000× *g* for 15 min at 4 °C, and the supernatants were stored at –70 °C until use. The total protein concentration in the supernatants was determined, and 50μL of the total protein from each sample was pipetted into an assay for MDA (Cell Biolabs, San Diego, CA, USA) and melatonin using the ELISA kit (catalog no. STA-832, Cell Biolabs, San Diego, CA, USA). The samples were analyzed according to the manufacturer’s instructions [3,4].

### 4.3. Immunohistochemistry

Immunohistochemistry (n = 1) was performed on cryosections of the eye and cornea. Tissues were fixed in paraformaldehyde overnight and 6 μm sections from paraffin wax blocks, mounted on precoated glass slides, deparaffinized, and rehydrated. Then, sections incubated at 4 °C with mouse monoclonal anti-4HNE (20 μg/mL; catalog no. 009MHN-202P; JaICA, Shizuoka, Japan), -MT1 (1:50; catalog no. SC390328; Santa Cruz, Dallas, TX, USA), -MT2 (1:50, catalog no. OABF00337, Aviva, CA, USA), or -p62 antibody (1:50; catalog no. ab211324; Abcam, MA, USA) for 1 h. After washing, the samples were incubated with secondary antibodies, including Goat anti-rabbit IgG H&L (for MT1, MT2, and p62; 1:1000; catalog no. ab205718; Abcam, MA, USA) and mouse IgG kappa binding protein (for 4-HNE; catalog no. SC-516102; Santa Cruz Dallas, TX, USA) for 1 h in the dark at room temperature, followed by three washes with phosphate-buffered saline. Sections were then counterstained with 4′,6-diamidino-2-phenylindole (DAPI; catalog no. H-1200; Vector, Burlingame, CA) for 5 min. Digital images of the representative areas of the cornea and conjunctiva were captured using a Leica upright microscope (DM2500; Leica Microsystems). The number of positively stained cells per 100 μm was counted.

### 4.4. Western Blotting

The expression of MT1, MT2, LC3-II, p62, BAX, and cleaved caspase 3 proteins was determined by Western blotting (n = 3). Proteins were extracted from the corneal tissues using a lysis buffer (M-PER; Pierce Biotechnology, Rockford, IL, USA) with a protease inhibitor cocktail. Lysates were centrifuged at 15,000 rpm for 10 min at 4 °C. The proteins (20 μg) in the samples were separated by 12% SDS-PAGE and were transferred to polyvinylidene difluoride membranes. The blots were then washed with TBST (10 mM Tris-HCl [pH 7.6], 150 mM NaCl, 0.05% Tween-20), blocked with 5% skim milk in TBST for 1 h, and incubated overnight at room temperature with primary antibodies, including mouse anti-MT1 (catalog no. sc-390328, Santa Cruz, Dallas, TX, USA), rabbit anti-MT2 (catalog no. ab203346, Abcam, MA, USA), rabbit anti-LC3-II (catalog no. ab192890), rabbit anti-p62 (catalog no. ab109012), mouse anti-BAX (catalog no. ab216494), and rabbit anti-cleaved caspase 3 (catalog no. ab214430). After incubation with secondary antibodies, immunoreactive bands were visualized using an enhanced chemiluminescence system (ECL Blotting Analysis System; Amersham, Arlington Heights, IL, USA). The data were analyzed via densitometry (Alliance MINI HD9; UVItec Ltd., Cambridge, UK). β-Actin was used as an internal control.

### 4.5. Ocular Surface Parameters

To investigate corneal epithelial injury, the CFS scores were measured by slit lamp bio-microscopy (magnification, ×16; BQ-900; Haag-Streit, Switzerland) under cobalt blue light. One microliter of 1% sodium fluorescein was instilled into the inferior conjunctival sac of each mouse using a micropipette. Ninety seconds later, punctate staining of the corneal surface was evaluated in a masked fashion. Each cornea was divided into four quadrants that were scored individually. CFS was calculated using a four-point scale: 0, absent; 1, slightly punctate staining <30 spots; 2, punctate staining >30 spots, but not diffuse; 3, severe diffuse staining but no positive plaque; and 4, positive fluorescein plaque. Four scores were added to generate a final grade (possible total of 16 points) [3].

### 4.6. Statistical Analysis

The Statistical Package for Social Sciences (SPSS, version 18.0, Chicago, IL, USA) was used for the statistical analyses. The results are presented as mean ± standard deviation (SD). Data were analyzed using one-way analysis of variance (ANOVA) with Dunnett’s T3 analysis; *p* < 0.05 was considered statistically significant.

## Figures and Tables

**Figure 1 ijms-23-11341-f001:**
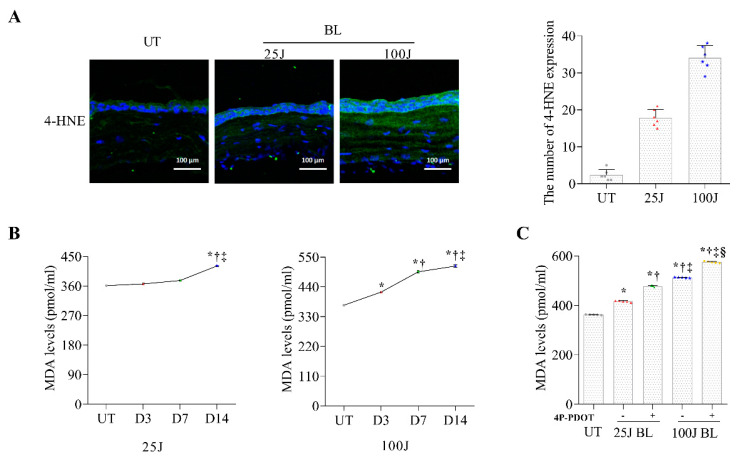
Showing representative photograph and the mean number of 4-HNE expression by IHC (**A**) in the cornea of the UT, 25 J BL, and 100J BL groups, after 14 days of BL exposure. ELISA for MDA (**B**) in the cornea of the UT, 25 J BL, 25J BL+4P-PDOT, 100 J BL, and 100 J BL+4P-PDOT groups. (**B**) * *p* < 0.05 vs. UT, ^†^
*p* < 0.05 vs. D3, ^‡^
*p* < 0.05 vs. D7; (C) * *p* < 0.05 vs. UT, ^†^
*p* < 0.05 vs. 25J BL, ^‡^
*p* < 0.05 vs. 25J BL+4P-PDOT, ^§^
*p* < 0.05 vs. 100 J BL; UT, untreated; BL, blue light.

**Figure 2 ijms-23-11341-f002:**
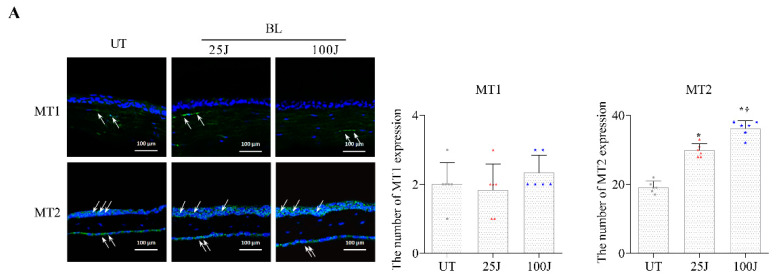
Expression of MT1 and MT2 by IHC (**A**) and Western blotting (**B**) in the cornea of the UT, 25 J BL, and 100 J BL groups, after 14 days of BL exposure. * *p* < 0.05 vs. UT, ^†^
*p* < 0.05 vs. D3, UT, untreated; BL, blue light.

**Figure 3 ijms-23-11341-f003:**
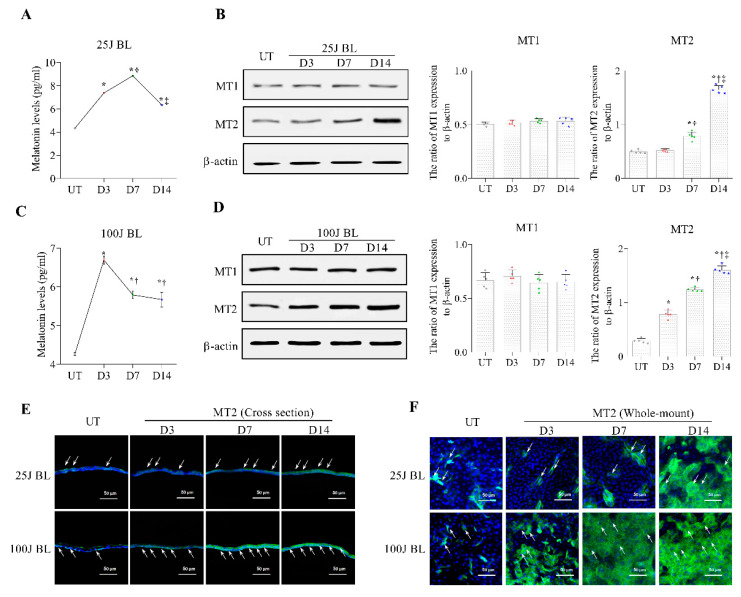
ELISA for melatonin levels (**A**,**C**) in the cornea of the UT, 25J BL, and 100J BL groups, after 14 days of BL exposure. Western blotting (**B**,**D**) for the expression of MT1 and MT2 in the cornea and IHC (**E**,**F**) for MT2 expression at days 3, 7, and 14. * *p* < 0.05 vs. UT, ^†^
*p* < 0.05 vs. D3, ^‡^
*p* < 0.05 vs. D7; UT, untreated; BL, blue light.

**Figure 4 ijms-23-11341-f004:**
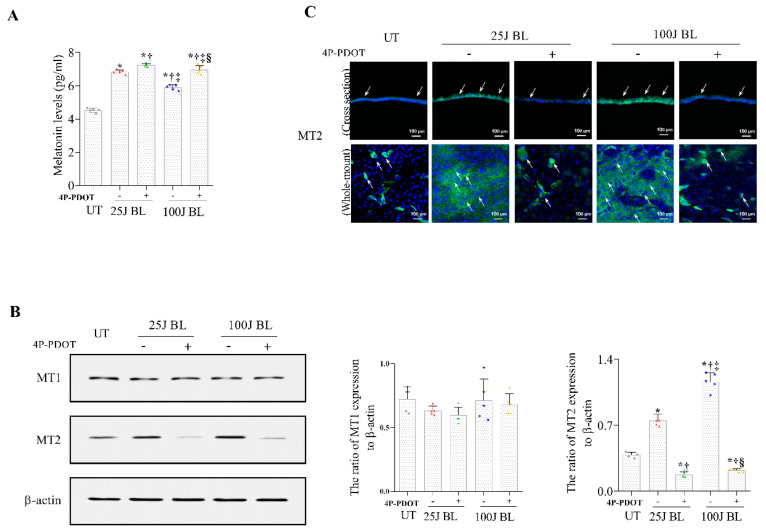
ELISA for melatonin levels (**A**) and Western blotting (**B**) for the expression of MT1 and MT2 in the cornea of the UT, 25 J BL, 25 J BL+4P-PDOT, 100 J BL, and 100 J BL+4P-PDOT groups, after 14 days of BL exposure. IHC (**C**) for MT2 expression in the mouse cornea treated or not treated with MT2 antagonist at day 14 of BL induction. * *p* < 0.05 vs. UT, ^†^
*p* < 0.05 vs. 25J BL, ^‡^
*p* < 0.05 vs. 25J BL+4P-PDOT, ^§^
*p* < 0.05 vs. 100 J BL; UT, untreated; BL, blue light.

**Figure 5 ijms-23-11341-f005:**
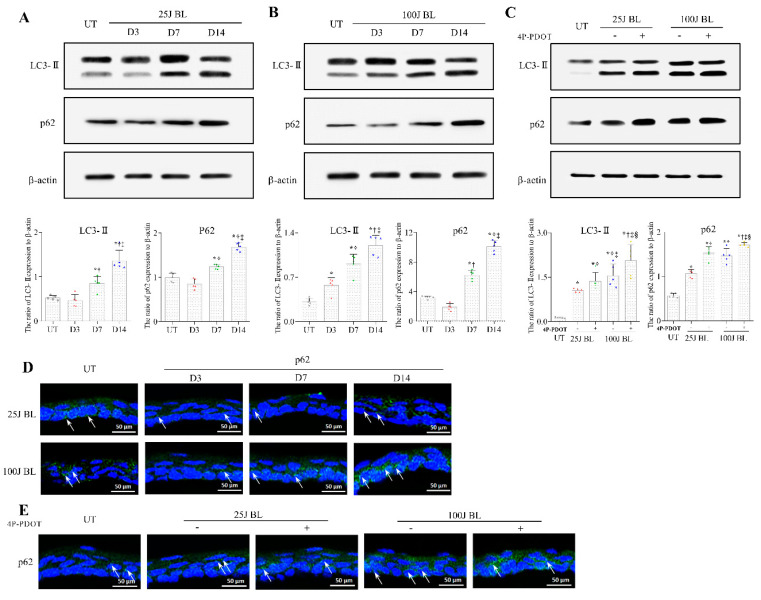
Western blotting (**A**–**C**) for the expression of LC3-Ⅱ and p62 and IHC (**D**,**E**) for p62 expression in the cornea of the UT, 25J BL, 25J BL+4P-PDOT, 100J BL, and 100J BL+4P-PDOT groups after 14 days of BL exposure. (**A**,**B**) * *p* < 0.05 vs. UT, ^†^
*p* < 0.05 vs. D3, ^‡^
*p* < 0.05 vs. D7; (**C**) * *p* < 0.05 vs. UT, ^†^
*p* < 0.05 vs. 25J BL, ^‡^
*p* < 0.05 vs. 25J BL+4P-PDOT, ^§^
*p* < 0.05 vs. 100J BL; UT, untreated; BL, blue light.

**Figure 6 ijms-23-11341-f006:**
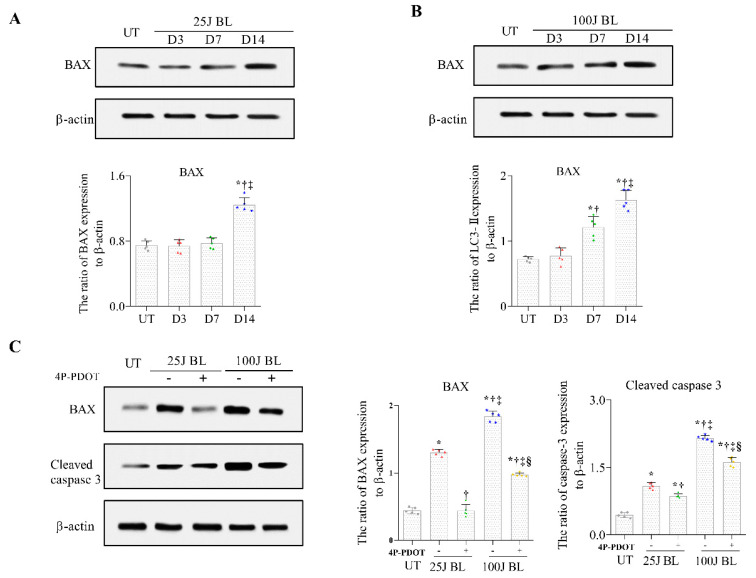
Western blotting (**A**,**B**) for BAX expression in the cornea of the UT, 25J BL, and 100J BL groups, on 3, 7, and 14 days of BL exposure. The expression of BAX and cleaved caspase-3 in the cornea of the UT, 25J BL, 25J BL+4P-PDOT, 100J BL, and 100J BL+4P-PDOT groups using Western blotting, after 14 days of BL exposure. (**A**,**B**) * *p* < 0.05 vs. UT, ^†^
*p* < 0.05 vs. D3, ^‡^
*p* < 0.05 vs. D7; (**C**) * *p* < 0.05 vs. UT, ^†^
*p* < 0.05 vs. 25J BL, ^‡^
*p* < 0.05 vs. 25J BL+4P-PDOT, ^§^
*p* < 0.05 vs. 100J BL; UT, untreated; BL, blue light.

**Figure 7 ijms-23-11341-f007:**
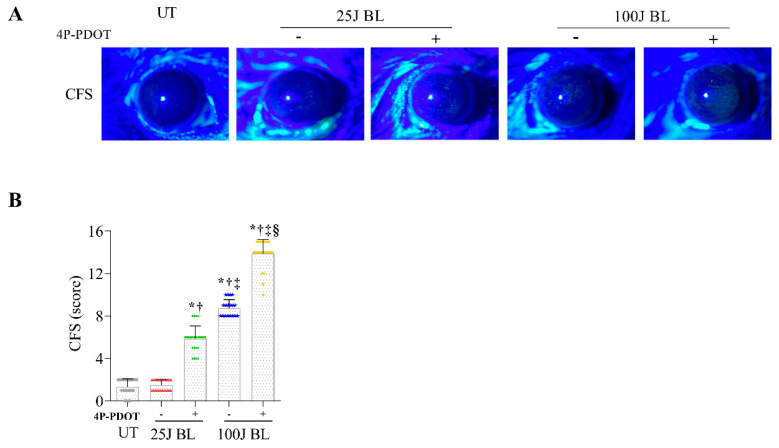
Representative 16× magnified photograph (**A**) and the mean corneal fluorescein staining scores (**B**) in the cornea of the UT, 25J BL, 25J BL+4P-PDOT, 100J BL, and 100J BL+4P-PDOT groups, after 14 days of BL exposure. * *p* < 0.05 vs. UT, ^†^
*p* < 0.05 vs. 25J BL, ^‡^
*p* < 0.05 vs. 25J BL+4P-PDOT, ^§^
*p* < 0.05 vs. 100J BL; UT, untreated; BL, blue light.

**Figure 8 ijms-23-11341-f008:**
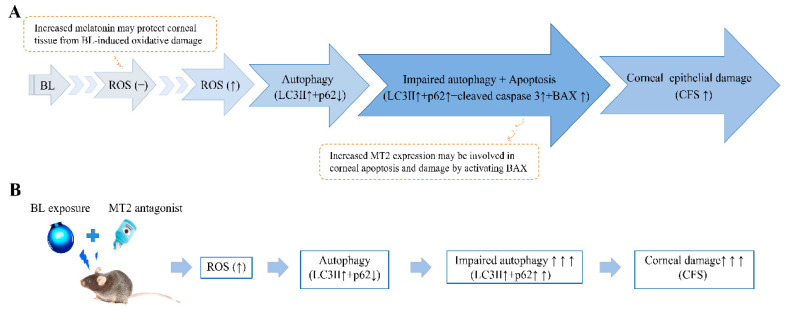
Showing the effects of BL exposure (**A**) and the role of MT2 antagonist (**B**) on mouse corneal epithelial damage.

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
