# Peer review of "Melatonin Type 2 Receptor Activation Regulates Blue Light Exposure-Induced Mouse Corneal Epithelial Damage by Modulating Impaired Autophagy and Apoptosis"

_ijms, 2022, doi:10.3390/ijms231911341_

Round 1
Reviewer 1 Report
The manuscript entitled "Melatonin Type 2 Receptor Activation Regulates Blue Light Exposure-induced Mouse Corneal Epithelial Damage by Modulating Impaired Autophagy and Apoptosis" is based on the role of melatonin receptors in the cascade of events involved in corneal epithelium damage. This study provides objective measurements and reports possible physiopathological pathways involved in the process of corneal damage and repair that are currently not completely known.
The topic adds to the current literature and is of clinical interest, especially considering the rarity of these types of experiments regarding this specific topic. The study plan and experiments are well designed. The protective role of melatonin in various organs and tissues in promoting homeostasis and reducing antioxidants is known, however, rather quite limited when dealing with ocular manifestations. The effects of blue light are becoming of great clinical interest in the era of computers and smartphones, thus of clinical pertinent interest. The results provide possible physiopathological mechanisms and pathways that can enrich our current knowledge in understanding the molecular and structural processes involved in corneal damage and repair.
I have several minor comments. Mention should be made in the Abstract section that MT1/2 are receptors of melatonin. The Discussion section should include possible future studies in this field (i.e. similar testing in mouse-damaged corneas induced by laser or chemical burn to study the effects of corneal epithelium and stromal repair treated with MT2 antagonist). Minor English revisions could improve the flow. A simple flow chart showing possible mechanisms, receptors, and molecular pathways involved in corneal damage and repair can be included to summarize and enhance the understanding of these various factors described throughout the text.
Author Response
Dear reviewer,
Thank you for your comments, and please see the attachment.

Reviewer 2 Report
This authors manuscript is describing the effects BL have on MT2 recpetor activation, as well as ROS production and Autophagy.
-Introduction should explain MDA and 4-HNE as oxidative stress markers and why they are used in this study to measure Oxidative stress.
-Spell out UT in first usage. I assumed it meant untreated.
- Figure 1 doesn't have a good description of what panel (C) is in the legend. I think it got left out of the legend text somewhere. There also needs to be an explanation of the Y-axis in figure 1C. Is it normalized to UT group? Is this set to 1? This needs to be explained, otherwise the amount of pmol/ml of MDA doesn't match up with figures 1B for similar groups.
- Figure 1A doesn't have indicators of what each fluorescent marker is telling, just that it is IHC for 4-HNE.
-Figure 2 western blot looks like the B-actin band in the UT lane is less than the others. When looking at the uncut Western Blot image, it is even more evident. This may account for the lower UT MT2 densitometry? It's hard to tell by looking, but I would like to see the densitometry of the B-Actin bands as well, to ensure the loading is the
-Figure legend 3 needs to be fixed. Some spelling errors, but also MT1 is not shown in IHC panels E and F, even though the legend says it is. What is the purpose and explanation for the difference between cross section vs. whole mount? No discussion in the results section of these. What analysis besides visual was done for E and F. It appears that there are much more MT2 in day 3 with 100J BL in both cross section and whole mount. This is a little contrary to what is seen in the western blots in previous panels.
- Figure 4B shows less densitometry for MT2 when antagonist incubated. The labeled on Y-axis reads "Intensity of MT2 expression". The word expression leads me to think of mRNA levels and that somehow the MT2 antagonist is causing a downregulation of MT2 expression. Is this the case? or is it due to cellular degradation? (the later text suggests the second option). I think the text needs to be a little more clear as to what is happening here.
Section 2.4 starts out with "As a result of evaluating the change in BAX expression after exposure to BL, similar to the change in MT2 expression, mice exposed to BL at 25 J showed increased BAX ex-pression in the cornea on day 14." I am confused since BAX has not been brought up yet to this point, so the first sentence is very misleading.
section 2.5 need to spell of CFS first time it is in text here.
The discussion is a little confusing to follow. I think what may really help in describing all of the data throughout the study is to have a sentence or two at the end of each results section describing the major impact of each result. It took me quite a few reads to put together all of the effects BL is having on the MT2. A cartoon or diagram of what all is being effected with the administration of BL would be extremely helpful and I think would really add to the quality of this paper. The results are interesting and I think a little more clarity in their description is needed to have a clear understanding of what they mean.
I also feel that the introduction should be better describing all of the markers and proteins that will be explored throughout the manuscript.
Author Response

(The authors gave the same response as above.)

Round 2
Reviewer 2 Report
All concerns were adressed.
Author Response
Thanks for your review.